# Future Climate Effects on Yield and Mortality of Conventional versus Modified Oil Palm in SE Asia

**DOI:** 10.3390/plants12122236

**Published:** 2023-06-07

**Authors:** Robert Russell Monteith Paterson

**Affiliations:** 1Department of Biological Engineering, Gualtar Campus, University of Minho, 4710-057 Braga, Portugal; russell.paterson@deb.uminho.pt; 2Department of Plant Protection, Faculty of Agriculture, Universiti Putra Malaysia, Serdang 43400, Selangor, Malaysia

**Keywords:** *Elaeis guineensis*, Indonesia, Malaysia, Papua New Guinea, climate change, temperature

## Abstract

Palm oil is a very important commodity which will be required well into the future. However, the consequences of growing oil palm (OP) are often detrimental to the environment and contribute to climate change. On the other hand, climate change stress will decrease the production of palm oil by causing mortality and ill health of OP, as well as reducing yields. Genetically modified OP (mOP) may be produced in the future to resist climate change stress, although it will take a long time to develop and introduce, if they are successfully produced at all. It is crucial to understand the benefits mOP may bring for resisting climate change and increasing the sustainability of the palm oil industry. This paper employs modeling of suitable climate for OP using the CLIMEX program in (a) Indonesia and Malaysia, which are the first and second largest growers of OP respectively, and (b) Thailand and Papua New Guinea, which are much smaller growers. It is useful to compare these countries in terms of future palm oil production and what benefits planting mOP may bring. Uniquely, narrative models are used in the current paper to determine how climate change will affect yields of conventional OP and mOP. The effect of climate change on the mortality of mOP is also determined for the first time. The gains from using mOP were moderate, but substantial, if compared to the current production of other continents or countries. This was especially the case for Indonesia and Malaysia. The development of mOP requires a realistic appreciation of what benefits may accrue.

## 1. Introduction

Climate change is having, in some cases, catastrophic effects on human activities. These will become even worse in the future, according to existing trends. These detrimental effects have been recognized, inter alia, at the recent meeting of the Conference of the Parties (COP) in Glasgow, Scotland [1]. The issue is taken extremely seriously by governments, scientists, and the general public. New technologies to combat the impacts are required, with the most important action being a radical reduction in CO_2_ emissions. The consequences of such climate stress are often assessed by modeling techniques, which frequently involve computer and narrative models.

A major issue is the effect of climate change on crop production, but there is limited information on tropical plants, including crops [2]. One of the most important is oil palm (OP), which produces palm oil and is included in a vast variety of commodities. For example, it is (a) present in ca. 50% of supermarket products, (b) used as biodiesel and (c) employed in pharmaceuticals.

OP expansion often results in deforestation [3,4,5,6,7], and indeed, COP 26 has declared the intention to end deforestation, consequently reducing climate change [8]. Furthermore, ecosystem function decreased greatly upon the introduction of OP plantations, and the effects of climate change on OP will increase social problems, particularly in producing regions [9].

CLIMEX computer modeling is essential in understanding the impacts on species distribution [10] due to climate change [11]. However, the *in situ* adaptation of crops involving the action of humans is possible [12], in contrast to novel geographical distributions of crops (i.e., *ex situ*) [13]. *In situ* adaption to climate change has been studied and includes the use of empty fruit bunches for soil remediation [14]; arbuscular mycorrhizal fungi [15]; and interplanting with other crops [16].

Another possibility is the creation of modified crops more suited to future climate stress which is being researched for OP, although it is at the early stages [17,18,19]. This technology is not straightforward [20], and genetically breeding mOP for climate change requires multidisciplinary and collaborative research at high levels of expertise. One problem is selecting for complete disease resistance, rather than tolerance, leads to high selection pressures for new variants of the pest/pathogen that can overcome the resistance in the crop. A long time will pass before results will translate into OP available for planting on a commercial basis. In addition, it is impossible to accurately determine what aspect of climate change to focus on for resistant cultivar development; for example, one resistant to desiccation stress may be sensitive to high temperatures [20].

Heat stress tolerance is complex and controlled by many genes. Studies have pointed to several sophisticated mechanisms for dealing with the stress of high temperatures, including (a) hormonal signaling pathways for sensing heat stimuli and acquiring tolerance to heat stress, (b) maintaining membrane integrity, (c) production of heat shock proteins, (d) removal of reactive oxygen species, (e) assembly of antioxidants, (f) accumulation of compatible solutes, and (g) modified gene expression to enable relevant changes. The manipulation of multiple genes responsible for thermotolerance and the exploration of their high expressions greatly impacts their potential application [21], consequently making this a difficult field of research.

A focus for the production of mOP is the *MYB* gene family [18], which has vital roles in secondary metabolite regulation, abiotic stress responses, and hormone signal transduction. MYB plant proteins have a highly conserved DNA-binding domain of 51~52 amino acid residues at the N-terminal region, with one to four imperfect repeat sequences [18]. *MYB* genes are known to regulate downstream genes at the transcriptional and post-transcriptional levels under abiotic stress conditions [22]. Finding the candidate genes and dissecting signaling mechanisms in abiotic stress responses is a key determinant in the development of abiotic stress-tolerant OP [23]. Overall, the manipulation of genes responsible for thermotolerance and investigating their high expression, is relevant to their usefulness in developing mOP. The response and tolerance to heat stress at the cellular, organelle, and whole-plant levels to enhance thermotolerance in oilseed crops is now understood, and can help improve breeding programs by greatly enhancing the efforts to establish heat stress tolerance in OP.

Furthermore, two OP progenies of (a) Tanzanian and (b) Nigerian origin, which were (a) adapted and (b) sensitive to drought stress, respectively, were studied. The Tanzanian progeny had more stable sap flow, greater water potential, better stomatal closure, and a lower proline content than the Nigerian progeny, and hundreds of potentially useful SNPs were detected. The study found that 9.3% of the SNPs had different alleles between progenies. Two SNP markers were located in the aquaporin NIP1-1 gene, related to water translocation between cells. One SNP marker was located in the glutamate receptor gene, which is related to glutamate release and proline biosynthesis [24]. Furthermore, expression data for all transcripts in both progenies demonstrated that numerous genes were expressed differentially, such as genes involved in primary hormone responses, DNA replication and repair, chromatin remodeling, and RNA-mediated DNA methylation. These gene expression data are a valuable resource for oil palm genetics [25].

The extent of climate change and its effects, are often determined by creating scenarios as distinct from predictions, which are impossible to make because the future cannot be predicted accurately. The scenarios are intended to be useful for, for example, planning purposes. They involve computer-generated models and are often combined with narrative models [26]. Novel uses of future climate maps to obtain granularly-detailed information have been found [27,28]. A qualitative estimation of the effect of the future climate on the cultivation of OP in Malaysia, and determinations of the incidence of basal stem rot (BSR), a serious fungal disease of OP, were made [29]. The scenarios were made quantitative for these parameters in Sumatra, Indonesia [30]. The same procedure was extended to Kalimantan and alternative SE Asian countries, including Thailand and Papua New Guinea (PNG). These assessments were conducted for OP mortality, BSR [31] and (a) bud rot disease caused by *Phytophthora palmivora* in South America compared to Malaysia and Indonesia [32] and (b) Fusarium wilt in Africa extrapolated to Malaysia and Indonesia [33]. Paterson [34] employed climate maps of Africa, South America, and SE Asia to determine longitudinal trends in the suitable climate for growing OP, which may indicate future refuges for OP. Longitudinal and latitudinal trends of OP refuges were determined for Nigeria [13].

Indonesia and Malaysia are the first and second highest growers, respectively, of OP, whereas Thailand and Papua New Guinea (PNG) grow much lower, but still significant, amounts (Table 1). The effect of future climate on growing OP in the countries has been determined [31,32]. The present study considers, for the first time, how future climate will affect the mortalities and yields of OP in these four countries until 2100, and compares this information to that obtained for mOP in a novel approach. The results can be employed to consider the potential benefits of creating mOP.

## 2. Results

OP mortalities were low, and similar in Malaysia and Indonesia (Figure 1). Much higher OP mortalities were determined in Thailand compared to Malaysia and Indonesia, with PNG being intermediate. OP mortalities remained at 0 until 2050 for Malaysia, Indonesia, and PNG, whereas they were 8% and 11% in 2030 and 2050, respectively, for OP in Thailand. The mortalities rose to 1%, 2%, 15%, and 23% for Malaysia, Indonesia, PNG, and Thailand, respectively, in 2070 and in 2100, these rose to 10%, 10%, 40%, and 70% for the same countries, respectively. The mortality for Malaysian mOP was 0.6% for 2070, which was a 0.4% decrease compared to cOP. This rose to 5% by 2100, a 5% decrease compared to cOP. For Thailand, the OP mortality rate was 15% for mOP in 2070, which was 8% lower than that for cOP: Mortality rose to 32% in 2100, which was 38% lower than that for cOP. In the case of Indonesia, mortality was 1.3%, which was 0.7% lower than cOP in 2070. This rose to 5% in 2100, which was 5% lower than that for cOP. For PNG, a mortality of 10% was recorded in 2070, which was 5% lower than cOP for this year. For 2100, this rose to 18%, which was 22% lower than that for cOP.

The yield of Malaysian palm oil did not change from current time to 2030 (Figure 2) and increased to 104% by 2050. The yield decreased to 88% and 49% by 2070 and 2100, respectively, for cOP. The yields for Malaysian mOP remained the same until 2070, reaching 52% in 2100, which was 3% higher than for cOP. In the case of Thailand, there was a precipitous decrease in yield from 2030 to 2100, with a 2100 value of zero for cOP and mOP. In 2070 the yield from mOP was 20% compared to 18% for cOP for Thailand. For Indonesia, there was an increase in yield to 104% by 2030, which was maintained until 2050. By 2070, the yield for cOP was 91% compared to 92% for mOP, and by 2100, the yield for cOP was 67% compared to 70% for mOP. In the case of PNG, there was a remarkable linear increase in yield to 115% by 2050, and then a rapid decrease to 55% by 2070 for cOP, compared to 58% for mOP. By 2100, the values were 10% and 17% for cOP and mOP, respectively, for PNG.

Table 2 indicates the number of OP plants remaining alive in 2070 and 2100 in Malaysia, Thailand, Indonesia, and PNG. Very large numbers of OP plants remained alive in 2070 in Indonesia and Malaysia at values of 891.80 × 10^6^ and 604.30 × 10^6^, respectively. The equivalent number for Thailand was lower by a factor of ca. 10, and even fewer plants were considered alive in PNG. By 2100, a large number of OP plants remained alive in Indonesia and Malaysia (819.00 × 10^6^ and 549.36 × 10^6^, respectively). The numbers of plants that remained alive at this time were much lower in Thailand and PNG. The largest number of OP plants saved by using mOP in 2070 was 7.22 × 10^6^ for Thailand, followed by Indonesia at 6.37 × 19^6^. Malaysia had 2.44 × 10^6^ and PNG had 1.99 × 10^6^ saved by using mOP in 2070. In general, many more OP plants were considered to have been saved by planting mOP in 2100. The largest number saved was in Indonesia at 46.50 × 10^6^, followed by Thailand at 34.31 × 10^6^. Malaysia and PNG had 30.52 × 10^6^ and 4.00 × 10^6^ OP plants saved, respectively, by planting mOP.

The yields in 2070 decreased slightly in Indonesia, from 29.83 × 10^6^ t yr^−1^ to 27.14 × 10^6^ t yr^−1^, which is still a high number (Table 2). Similarly, Malaysia experienced a decrease from 21 × 10^6^ to 18.48 × 10^6^ t yr^−1^. There was a larger decrease in yield in Thailand, from 1.75 × 10^6^ t yr^−1^ to 0.31 × 10^6^ t yr^−1^. PNG had an intermediate decrease from 0.56 × 10^6^ t yr^−1^ to 0.31 × 10^6^ t yr^−1^. Substantial yields remained for Indonesia and Malaysia until 2100, with Indonesia and Malaysia having 20.0 × 10^6^ and 10.0 × 10^6^ t yr^−1^, respectively. The yield was zero in Thailand in 2100, and the yield for PNG was 0.06 × 10^6^ t yr^−1^. 

Modest or no increases in yields were determined for the four countries using mOP in 2070 (Table 2). The largest increase in yield from using mOP was found for Indonesia, at 0.3 × 10^6^ t yr^−1^. There was no increase in yield due to the use of mOP in Malaysia in 2070. The increase in yield for Thailand was 0.04 × 10^6^ t yr^−1^, and that for PNG was 0.01 × 10^6^ t yr^−1^ in 2070. By 2100, the highest increase in yield from using mOP was found for Indonesia, at 0.89 × 10^6^ t yr^−1^, followed by Malaysia, at 0.63 × 10^6^ t yr^−1^. PNG had an increase of 0.04 × 10^6^ t yr^−1^ by 2100, and there was zero yield for Thailand at this time.

## 3. Discussion

The present paper concerns the effect of the future climate on yields and OP mortality in four SE Asian countries, and considers the benefits of planting mOP. The need for mOP is much greater in Thailand than Malaysia, Indonesia, and PNG because of the higher losses of OP in Thailand, even by 2030 (Figure 1). However, this will not be addressed under the current scenario, because mOP is not available until after 2050. The 10% loss of OP by 2100 in Malaysia and Indonesia represents a large number of OP, considering how many OP plants exist in these countries (Table 2), although the loss may be considered as undramatic and containable by the OP industry. It is reassuring that the two principal palm oil producers, Indonesia and Malaysia, did not show high losses. The number of OP plants saved in 2100 for Indonesia was 45.50 × 10^6^, which is approximately 50% of the total number of OP plants currently in Thailand and ca. 200% of the equivalent number in PNG (Table 2). Hence, it may be worthwhile for Indonesia to invest in mOP technology on this basis. The numbers saved in Malaysia were also high, reaching 30.52 × 10^6^ OP by 2100.

It is interesting that the number of OP plants saved in Thailand by 2070 was high, despite the number of plants at current time being almost a factor of 10 lower than Malaysia and Indonesia. However, the yield of palm oil was zero by 2100 for cOP (Table 2), making climate change much more of a threat. By 2070, there was an increase in OP of 7.22 × 10^6^ and a moderate increase in yield in Thailand due to the introduction of mOP (Table 2), which may make the investment in mOP worthwhile. Small improvements in OP plants saved and the yields of palm oil were observed for PNG, making it uncertain the gains would be worthwhile.

There were increases in yields until 2050 for Malaysia, Indonesia, and especially PNG which were related to an increase in the suitability of the climate [31] (Figure 2). Hence, there would not be a requirement for mOP initially, reducing the requirement for mOP. The decreases in yields of palm oil were small in Malaysia and Indonesia by 2070, and increases were larger for Thailand and, to a lesser extent, PNG (Table 2). By 2100, the yield was zero in Thailand, as mentioned previously, which would be catastrophic for the industry if it were to occur in reality. In PNG, the yield was severely reduced to 0.06 × 10^6^ t yr^−1^ for cOP, which would be highly detrimental to the OP industry.

Furthermore, the yields in Malaysia and Indonesia by 2100 were ca. 50% of the yields in current time for Thailand and PNG. The increases in yield due to planting mOP may be worthwhile at 0.63 × 10^6^ and 0.89 × 10^6^ t yr^−1^ for Malaysia and Indonesia, respectively, which are greater than the yield for PNG in current time.

The data for the countries discussed herein (Table 2) can be compared to information on other continents and countries (Table 1). The numbers of OP plants saved in Thailand and Indonesia in 2070 were more than half of the total number of OP plants currently in India (Table 1), indicating that the numbers are significant. However, the yield improvement for Thailand in 2070 was considerably less than 50% of the current yield of India, and, consequently, the introduction of mOP in Thailand may not be particularly beneficial. The yield improvement for Indonesia in 2070 was approximately three times that that for India, indicating the high value of introducing mOP in Indonesia. The numbers of OP plants saved by 2100 in each of Malaysia, Indonesia, and Thailand were 3 to 4 times higher than the total number of OP plants currently in India and were similar to the total number in Central America (Table 1), again indicating the value of introducing mOP in the long term. The yield increases due to employing mOP was approximately 6 to 8 times higher than the total yield from India currently, and approximately half that of Central America in the case of Malaysia and 75% in the case of Indonesia. Again, this clearly indicates the potential benefits of planting mOP in these countries. The number of OP plants saved in Malaysia by 2070 was less significant, and developing mOP may not be worthwhile initially. In addition, there was no increase in yield. By 2100, considerable increases were seen for both parameters in Malaysia, as mentioned previously, and so the situation is more nuanced.

Estimations equivalent to those for the four countries considered in the present report can be derived to a lesser extent for other countries. The Philippines [31] and Cameroon [33] were considered to have benign future climates for the growth of OP, making the situation somewhat similar to those for Malaysia and Indonesia in the current report. Myanmar [31], Ghana [33], Nigeria [33], Brazil [32], and Columbia [32] had detrimental future climates, and it is possible that the future OP data would be similar to that of Thailand in the current report. Ecuador [32] may be in an intermediate situation, as the future climate is most similar to PNG. Calculations similar to those made for the four countries detailed in this paper will be required for these other countries.

In addition, Paterson [34] reported longitudinal trends in climates suitable for creating potential OP refuges in SE Asia that included the countries considered in the current report. Clearly, another procedure for mitigating the effects of unsuitable future climate would be utilizing the longitudinal trend by planting in regions less affected by climate change. The paper demonstrated that Sarawak and Sabah (i.e. Island Malaysia) showed an increase in suitable climate in 2050; thus, mOP may not be required. For Peninsular Malaysia, there was no change in suitable climate in 2050; again, mOP may not be required. Of course, mOP may not be available in 2050 as considered herein, which would give Malaysia additional time to develop mOP. Thailand had the largest and most dramatic decrease in suitable climate by 2050 [34], and effective mOP would be highly desirable in this case. The country had a climate with a very low level of suitability by 2100, emphasizing the need for mOP. All other countries or regions were less detrimentally affected than Thailand by 2050, and these may become alternative regions for introducing plantations.

By 2100, Peninsular Malaysia showed a much higher decrease in climate suitability [34], and the introduction of mOP would be critically important in this region. However, Sarawak and Sabah showed only a small decrease in suitable climate by 2100, and so mOP planting would be less critical. OP cultivation could be moved from within Malaysia to Island Malaysia. Most of the regions in Indonesia did not show a significant loss of suitable climate by 2050, but did show substantial losses by 2100. Sulawesi had the smallest loss, at −20%. For PNG, there was an increase in suitable climate by 2050, and so mOP would not be advantageous; however, by 2100, the loss was significant, at 55% [34].

High levels of resources will be required for the creation of mOP suitable for introduction into plantations. Particularly efficient OP companies could conceivably develop and plant mOP earlier than suggested herein, and/or could achieve a higher planting rate, resulting in lower mortality and greater yield. The reverse is true in the less efficient plantations. For example, smallholder farmers may not have access to mOP and would experience severe losses, perhaps making their businesses unviable. The OP and yield improvements described in the current paper may be considered to be worthwhile, especially if palm and other vegetable oils become less available because of the detrimental effects of future climate change on oil-producing crops.

## 4. Materials and Methods

The models described in Paterson et al. [27,28] provided scenarios for growing OP under climate change conditions. Paterson et al. [28] provided information from maps of Malaysia and Indonesia for the current time, 2050, and 2100. Paterson et al. [27] provided maps of these countries for the current time, 2030, 2070, and 2100. The models described in Paterson et al. [28] provided climate scenarios suitable for growing OP under climate change conditions for the current time, 2050, and 2100 in Thailand and PNG [31]. The maps were magnified on a computer screen to focus on these countries using the standard magnification facility. The percentages of suitable climates were assessed visually to provide percentage suitabilities determined from the designated colors of the maps, where each color represented a particular degree of climate suitability for growing OP.

The distribution model for OP under different climate scenarios was developed using CLIMEX for Windows, Version 347 (Hearne Scientific Software Pty Ltd., Melbourne, Australia, 2007), and climate data and climate change scenarios were assessed using CliMond 10’gridded climate data. The potential future climate was characterized using A1B and A2 SRES scenarios [27], which are available from the CliMond dataset. The fitting of CLIMEX Parameters employed the Global Biodiversity Information Facility. Information on the global distribution of OP was used in parameter fitting, and 124 records were used. SE Asian distribution data were reserved for the validation of the model. The OP distribution was determined by the Global Biodiversity Information Facility (GBIF) (http://www.gbif.org/, accessed 9 November 2015) and additional literature on the species in CAB Direct (http://www.cabdirect.org/web/about.html, accessed 9 October 2015), and the basis was formed for the collection of data regarding the distribution *Elaeis guineensis* in reference [28], with 2465 records utilized in fitting the parameters. CLIMEX was used along with the A2 Special Report on Emissions Scenarios (SRES) scenario. A mechanistic niche model using CLIMEX software supported ecological research incorporating the modeling of species’ potential distributions under differing climate scenarios, and assumed that climate was the paramount determining factor of plant and poikilothermal animal distributions. CLIMEX outputted categorized areas according to climate, designating highly suitable climates, suitable climates, marginal climates, and unsuitable climates. This was based on other studies conducted through CLIMEX.

The suitable climate for growing OP information was used to determine OP mortality by identifying where large percentages of unsuitable and marginal climates were likely to cause high mortality, and vice versa. In addition, reductions in highly suitable and/or suitable climates would not have such a significant effect on OP mortality [31,32]. The data on mortalities in Thailand and PNG were extrapolated from the relevant mortality data in reference [31] for 2030 and 2070.

For the scenario used in this paper, mOP was considered not to become available until after 2050 (10% every 5 years) due to the complexity of producing such plants. To calculate the mortalities for modified OP, the following was employed:

The mortality of mOP was determined by taking the mortality of cOP and multiplying it by the percentage introduced since the previous date (i.e., 40/100 = 0.4 for 2070 and 60/100 = 0.6 for 2100). Ninety percent of this figure was taken as the efficiency factor to compensate for the probability that mOP would not be 100% effective at combating climate change. This gave the percentage of OP which was saved. This figure was subtracted from the mortality for cOP to provide the mortality for mOP.

The yield was calculated using the following equation:Yield (%) = G × f

G = percentage of OP plants remaining alive for each year; f = climate change factor for the increasingly detrimental climate (Table 3).

Other calculations

Number of OP plants planted in the current time was determined from the following equation:No. OP = a × b 

a = number of OP plants planted per ha; b = total ha planted.

The number of OP plants planted per acre was taken to be 140, as a reasonable estimation [35] for both countries.

The total area planted for each country was determined from reference [35].

The total yield per year (Y yr^−1^) for each country was determined from:Y = p × q 

Y = yield yr^−1^; p = yield (ha^−1^ yr^−1^) [35]; q = area planted (ha) [35].

## 5. Conclusions

The future availability of mOP is not a given, nor a panacea, for problems relating to climate change in the OP industry. There will not be a complete reversal of OP mortality nor a restoration of palm oil yields. This may be due to the somewhat prosaic factors of the time taken to create mOP and the rate of planting. The efficiency of the mOP at combating climate change is also a factor. Future climate will not affect the mortality or the yields of OP equally in different countries, as Thailand will be badly affected and Malaysia and Indonesia much less so. There is a need to assess the value of developing mOP further, and which will require a separate consideration for each country. The benefits may not be obvious, but could be considered worthwhile. This is especially true when compared to the current palm oil production of other countries or continents, as the improvements described in the current paper are of the same order as some of these other regions.

Furthermore, the economic value of palm oil may increase greatly in the future as it becomes scarce due to climate change, making small volumes of palm oil worth producing. However, it is likely that the consequences of unabated climate change will be severe and the economies of countries will be dramatically changed to such an extent that markets will be different from those that apply at the present time.

## Figures and Tables

**Figure 1 plants-12-02236-f001:**
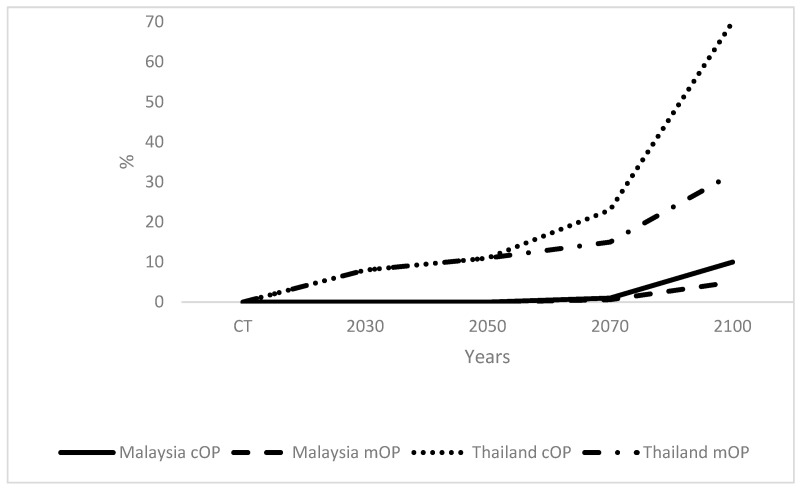
Oil palm mortalities of conventional oil palm (cOP) and modified oil palm (mOP) in Malaysia, Thailand, Indonesia, and Papua New Guinea (PNG) from current time (CT) to 2100.

**Figure 2 plants-12-02236-f002:**
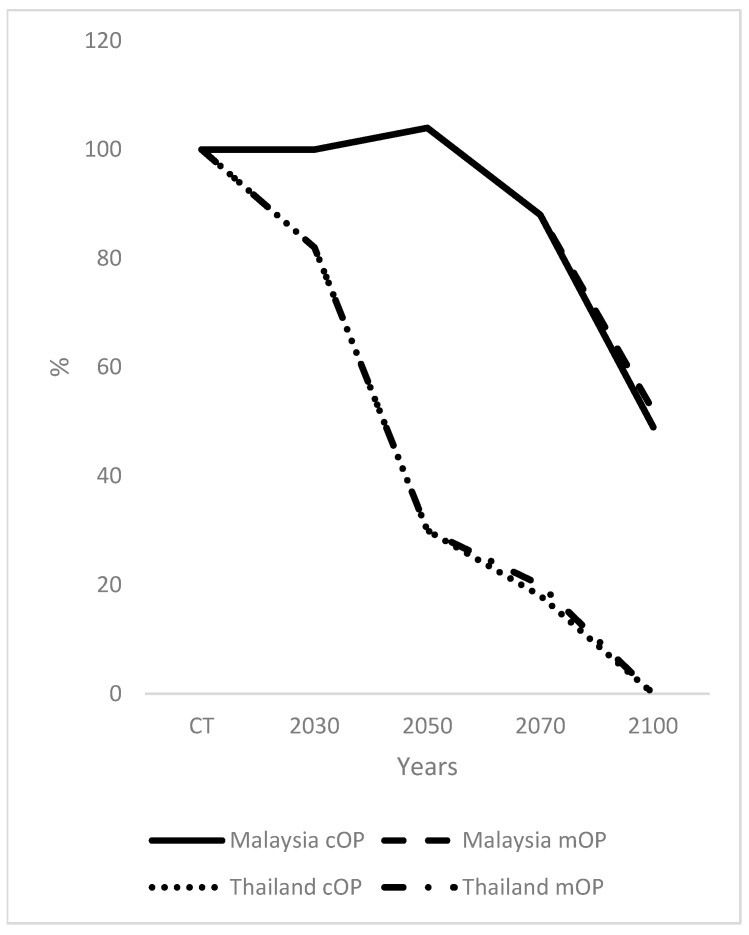
Yields of palm oil from Malaysia, Thailand, Indonesia, and Papua New Guinea (PNG), including conventional OP (cOP) and modified oil palm (mOP), from the current time (CT) to 2100.

**Table 1 plants-12-02236-t001:** Total area planted (ha), number of oil palm (OP) plants, and yields (t yr^−1^) in various continents and countries (×10^6^) [35].

	Hectares	Number of OP Plants	Yield
Africa	1.25	175	2.43
Central America	0.29	41	1.24
South America	0.71	99	2.19
India	0.08	11	0.10

**Table 2 plants-12-02236-t002:** Mortality of oil palm (×10^6^) in the future climate as a percentage of the total number in the current time (CT) in each country. Yields (t yr^−1^ × 10^6^) of palm and palm kernel oil from conventional oil palm (cOP) compared to modified oil palm (mOP) [35].

			Year		
	CT	2070		2100	
*Country*	*Alive*	*Alive*	*Saved*	*Alive*	*Saved*
Malaysia cOP/mOP	610.4	604.296/606.7376	2.4416	549.36/579.88	30.52
Thailand cOP/mOP	90.3	69.531/76.755	7.224	27.09/61.404	34.314
Indonesia	910	891.8/898.17	6.37	819/864.5	45.5
PNG	20	17/18	1	12/16	4
	*Yield*	*Yield*	*Increase*	*Yield*	*Increase*
Malaysia cOP/mOP	21.0	18.48/18.48	0.00	10.29/10.92	0.63
Thailand cOP/mOP	1.75	0.31/0.35	0.04	0.00	0.00
Indonesia	29.8	27.14/27.44	0.30	19.99/20.88	0.89
PNG	0.6	0.31/0.32	0.01	0.06/0.10	0.04

**Table 3 plants-12-02236-t003:** Climate change factors (f) were used to determine yields. ^a^ The factors for Thailand and Papua New Guinea for 2030 and 2070 were extrapolated from the graphs in [31], as they were not determined in the paper per se.

	Year			
Country	2030	2050	2070	2100
Indonesia	104	104	93	74
Malaysia	100	104	89	55
Thailand	89 ^a^	34	23 ^a^	0
Papua New Guinea	105 ^a^	115	65 ^a^	17

## Data Availability

Data are contained within the article.

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
