# Peer review of "Future Climate Effects on Yield and Mortality of Conventional versus Modified Oil Palm in SE Asia"

_plants, 2023, doi:10.3390/plants12122236_

Round 1
Reviewer 1 Report
Dear Authors,
This research is meaningful for pursuing agricultural benefits for the future.
However, this manuscript needs to be sufficiently ready for publication.
First, you have to follow the MDPI author instructions and formats.
Second, the methodology is of utmost importance in the climate change research field. Depending on the researcher's methodology, the results can be totally different. So, you have to describe the method you used as much as possible, like why the researcher chose this scenario. However, you just said in the manuscript to check the method from other research, and it is so unkind to the authors.
Please review your research with other reviewers' comments and submit it again.
Thank you.
Reviewer 2 Report
I have serious doubts about the use of modelling methods in genetically modified crops as I consider that there are no past data to extrapolate to future scenarios. I also don’t understand some figures and I also suggest the review of the manuscript by an expert in climatology.
Reviewer 3 Report
The article "Future Climate Stress on Yield and Mortality of Conventional 2 Versus Modified Oil Palm in SE Asia" takes into consideration how future climate will affect the yield of OP in the four countries until 2100. It compares this information to that of a scenario created for mOP for mortality and yields in another novel approach. Although I found the article very ambitious, I thought the idea was fresh, well-explained, and laid out.
Round 2
Reviewer 1 Report
The contents and subject of this manuscript look good and easily readable for other researchers.
However this manuscript needs to give the readers scientific findings.
And you just used the simple method for estimating the yields. Please add the extra contents in methods for describing the climate data what you used.
Please keep the Plant journal format (MDPI journal format). The order of contents is different. Please check the tittle and legends of figures (and tables) for the readability as well.
Thank you for submitting your paper in Plant journal.
Reviewer 2 Report
.
Round 3
Reviewer 1 Report
Thank you for submitting your paper to Plants journal.